# Glucose as a Potential Key to Fuel Inflammation in Rheumatoid Arthritis

**DOI:** 10.3390/nu14112349

**Published:** 2022-06-04

**Authors:** Kayo Masuko

**Affiliations:** 1Department of Internal Medicine, Akasaka Sanno Medical Center, Tokyo 107-8402, Japan; k_msk@mac.com; Tel.: +81-3-6230-3701; Fax: +81-3-6230-3702; 2Clinical Research Center, International University of Health and Welfare, Tokyo 107-8402, Japan

**Keywords:** rheumatoid arthritis, glucose, autoimmunity

## Abstract

Glucose is the most important source of energy and homeostasis. Recent investigations are clarifying that glucose metabolism might be altered in rheumatoid arthritis (RA), which would play a role in the inflammatory phenotype of rheumatoid synovial fibroblasts. It may also play a role in a variety of autoimmune diseases’ pathophysiology by modulating immune responses and modifying autoantigen expressions. The research into glucose and its metabolism could lead to a better understanding of how carbohydrates contribute to the occurrence and duration of RA and other autoimmune diseases.

## 1. Introduction

Monosaccharide glucose (C_6_H_12_O_6_) is the central component of carbohydrates and is an essential nutrient for the human health. To maintain a metabolic homeostasis, we must consume or obtain daily an adequate glucose amount biochemically. The glucose concentration in our serum (almost synonymous with blood sugar) is strictly controlled endocrinologically and biochemically in normal conditions. Furthermore, besides being a fuel of the energy-producing system in the body, recent studies are unveiling the effects of glucose in multiple facets of the immune responses. The author’s goal in this concise article is to introduce the recent topics concerning the glucose role in rheumatoid arthritis (RA), which is an autoimmune-related inflammatory joint disease.

## 2. Glucose Metabolism Shifts to Glycolysis in Rheumatoid Synovium

We can consume glucose from dietary intake as a form of carbohydrate or saccharide. Even in situations of starvation, the trials to maintain blood sugar would occur through the glycogen breakdown or the gluconeogenesis process.

Glucose enters the cells via membrane-associated carrier proteins such as glucose transporters (GLUT) and sodium-dependent glucose transporters (SGLT) [1]. Glucose is metabolized by the glycolytic pathway after it enters the cytoplasm, with hexokinase (HK) serving as the initial response of phosphorylation of glucose to glycose-6-phosphate (G6P). The glycolysis results in converting one molecule of glucose to two molecules of pyruvate, which yields two molecules of reduced nicotinamide adenine dinucleotide (NADH) and two adenosine triphosphate (ATP). This glycolysis process does not need oxygen and can occur in anaerobic conditions. In aerobic conditions, the pyruvate will be transported into mitochondria, converted to acetyl-CoA, and entered into the citric acid cycle.

After a series of enzymatic reactions, the pyruvate is converted into carbon dioxide (CO_2_), which produces NADH. Furthermore, it is then converted into ATP, which is the “energy currency,” via mitochondrial oxidative phosphorylation (OXPHOS). When the molecular oxygen supply is insufficient, the pyruvate is converted to lactate, regenerating NAD+ and producing less ATP. This anaerobic glycolysis can occur in cells that proliferate rapidly, such as tumor cells [2].

On the other hand, G6P can enter another metabolic pathway, i.e., the pentose phosphate pathway (PPP), which generates a pentose (ribose) and a coenzyme nicotinamide adenine dinucleotide phosphate (NADPH). The NADPH is an essential reducing factor for fatty acid production; thus, the PPP is a crucial metabolic pathway to produce nucleic acids and fatty acids.

There is evidence that showed changed glucose metabolism in chronic inflammatory conditions such as in tumor cells [3,4,5]. As for RA, the pathological inflammation activates synovial fibroblasts in articular joints. In addition, the cells in patients with RA demonstrate metabolic alterations including a shift to glycolysis [3,4,6]. It has been shown that rheumatoid synovial fibroblasts (RASF) have an increased expression of the HK and GLUT-1 genes, which results in a more inflammatory phenotype of the fibroblasts [7,8]. The glycolytic shift in RASF is considered as a hypoxia consequence and an effect of a complex of inflammatory cytokines that includes the tumor necrosis factor (TNF) from the inflammatory cells and the activated helper T (Th) cells [6,9,10,11]. To support the finding, a glycolysis inhibitor reduced the inflammatory response, which is an aggressive phenotype of RASF [9].

On the other hand, in glucose metabolism, reactive species, including the reactive oxygen species (ROS), are generated mainly during the OXPHOS [12]. In hyperglycemia or diabetes, the ROS generation increases, while the antioxidant defense system deteriorates, resulting in diabetic complications of vascular damage and inflammation [12]. The mechanisms of such inflammatory responses by the ROS include an increased formation of advanced glycation end-products (AGEs) and an activation of inflammatory signals such as protein kinase C and nuclear factor-kappa B (NF-κB), which would further activate the expressions of proinflammatory molecules [12]. Of note, it is difficult to prove that the molecules of glucose per se are detrimental to inflammation since chronic hyperglycemic states would accompany a plethora of metabolic disorders [13]. Nevertheless, the interactions between hyperglycemia, ROS, and inflammation have been documented in previous studies. In addition, the potential effects of antidiabetic drugs on inflammation have also been demonstrated [14].

The altered metabolism in RA may be modulated by disease-modifying antirheumatic drugs (DMARDs). As an in vitro finding, Janus kinase inhibitors, but not cytokine-specific biologic DMARDs, suppressed the Th-cell-stimulated glycolytic activity as well as its inflammatory phenotype in RASF. This showed the involvement of a broad range of cytokines but not of a single one [10]. It is suggested that HK is the target of the RASF-specific strategy for the RA-specific glycolysis regulation because HK is highly expressed in the RASF [8,15].

## 3. Glucose May Enhance Autoimmune Responses via Antigen Modification

The post-translational modifications of proteins, or further of nucleic acids or lipids, have been identified as an important factor in RA pathogenesis [16]. Such modifications include both enzymatic (e.g., phosphorylation, methylation, and ubiquitination) and nonenzymatic (e.g., glycation, oxidation/reduction, and acetylation) changes [16]. Several post-translationally modified proteins have already been identified as autoantigens in the RA sera [17]. For example, autoantibodies to post-translationally carbamylated or acetylated proteins have been discovered in the RA [17,18,19,20]. The citrullinated proteins are the representative example of such modified proteins because the antibodies against the citrullinated proteins (anticitrullinated protein antibodies or ACPA) are a well-established hallmark of RA diagnosis and disease assessment [18]. The ACPA is recognized as being disease-specific to RA. Furthermore, it is also known to be present in RA-prone individuals, years before the symptoms appear [20].

Glycation, an irreversible nonenzymatic attachment of excess glucose to biomolecules, is also a post-translational modification process that occurs when hyperglycemia persists, which yields AGEs. Besides diabetic condition, oxidative stress and inflammation during aging or chronic age-related inflammatory diseases also result in AGE formation, which would occur not only in protein but also in lipids or nucleic acids, thus impairing various physiological functions [21,22]. Elevated AGE levels have been reported in patients with RA [23].

Importantly, AGEs are reported to have an antigenic potential which would lead to induced autoimmunity [24,25]. Khan et al. [24] reported, using in vitro glycated serum albumin, that the human sera responded and produced the antibodies targeting the glycated protein. The antibody-producing response was higher in more aged subjects and in smokers. The authors hypothesized that glycation and AGEs could create “neo-epitopes” on blood proteins and that older people with a smoking history would be more likely to respond to the newly generated autoantigen. This finding could be significant because smoking, including passive smoking during childhood, has been identified as a risk factor for the RA occurrence and severity [26,27,28].

Therefore, the potential relationship connecting aging, post-translational modification like glycation, and autoimmunity may give a clue to understanding the RA pathogenesis. In this issue, Vitásek et al., hypothesized that, rather than RA causing increased protein glycation, individuals who were prone to glycation may be more vulnerable to developing RA. This hypothesis appeared intriguing and important to investigate [23].

## 4. Diabetes and Autoimmunity: Is There a Link through DPP-4?

The causative relationship between diabetes mellitus, both the type 1 (T1DM) and type 2 (T2DM), and RA has been widely documented [29,30,31]. Especially, the metabolic relationship between T2DM and RA is under active investigation since insulin resistance has been thought to bridge the two conditions [32,33]. Insulin resistance is a pathological condition in which the insulin does not sufficiently effect through the insulin receptors (IRs). The IRs are expressed ubiquitously on the cell surface of adipose tissue, muscle, and many other cells including the synovial cells and T lymphocytes under regulation by the inflammatory cytokines [34,35,36]. The insulin signaling via IRs is thought to play an important role in RA, both metabolically and immunologically [7,37,38]. For example, the insulin signaling via IRs regulates the development of regulatory T cells (Treg); in addition, hyperinsulinemia, which is an insulin resistance hallmark, may lead to Treg function suppression [34,35].

The incretins are gut-derived peptides that are secreted in response to oral glucose ingestion and enhance the insulin secretion from the pancreatic islet beta cells [39,40]. The incretins include glucagon-like peptide (GLP)-1 and gastric inhibitory polypeptide (GIP), which are degraded by a glycoprotein dipeptidyl peptidase (DPP)-4, also known as CD26 [41,42]. The DPP-4 inhibitors (DPP-4i) were developed to maintain incretin levels for the preservation of glucose homeostasis and are now widely used as antidiabetic drugs [41,42]. Besides glucose homeostasis, DPP-4 has immunomodulatory functions as the DPP-4/CD26 is expressed in a variety of immune cells [42,43,44]. There are reports showing the occurrence of rheumatic diseases after DPP-4i use in patients with diabetes [45,46,47,48] or an association of DPP-4 expression or its enzymatic activity with the disease activity of RA [44]. For example, Padron et al. reported a case report of a seronegative RA occurrence after taking sitagliptin [47]. The authors described how discontinuing sitagliptin, along with other therapeutic management, resulted in RA remission in this case.

However, in larger studies [49,50], the association between the DPP-4i use and the occurrence of clinical RA has been called into question. On the contrary, the DPP-4i use may be associated with a lower risk of incidence of RA in patients with diabetes [49,50]. The mechanism of the possible protective effect of the DPP-4i against RA is still unclear; however, it may be due to the suppression of inflammatory cytokines and cytokine receptor expression via CD26. In addition, the DPP-4i would decrease chemotaxis and T cell immunity that might trigger the autoimmune disease occurrence [49]. Nonetheless, we should be aware of the potential risk of DPP-4 inhibitors causing autoimmune diseases such as RA or bullous pemphigoid because there are case reports or incidents as mentioned above [51,52,53].

## 5. Modulation of the Gut and Oral Microbiome by Glucose

There are trillions of microbes colonizing the skin and mucosal surface in humans. Among the microbes, the commensal bacteria impact in the gastrointestinal tract in a variety of immune responses, and further in autoimmune arthritis, has attracted recent studies [54,55,56]. Firmicutes and Bacteroidetes are the most dominant bacterial phyla in the intestinal gut. However, there is considerable variation in the species levels among individuals [55]. In this issue, many reports documented a dominance of *Prevotella* spp., in particular of a taxon *Prevotella copri*, among the gut microbiota in recent-onset patients with RA or individuals with RA risk [56,57,58,59]. Thus, this suggests a potential link between gut microbiota and RA development. The precise mechanism of *Prevotella copri*’s possible arthritogenic mechanism is unknown; however, Prevotella spp. have been reported to activate responses via toll-like receptors and induced Th17 polarization, which lead to gut inflammation and the systemic dissemination of inflammatory mediators [59]. Furthermore, *P. copri* has been suggested to associate with insulin resistance as it could be seen as a predominant species in patients with diabetes and would induce insulin resistance [60,61]. As for RA, Maeda et al. [54] demonstrated, using an animal model, that *P. copri* stimulated autoreactive T cell responses. Interestingly, the gut microbiota with *P. copri* abundance could be modified by adherence to a specific dietary type such as the Mediterranean diet [62]. This might explain in part the reports describing the efficacy of the Mediterranean diet in RA [63].

The components of commensal microbiota in the human intestine are controlled by multiple factors, which include the dietary component as described above [64,65,66,67]. In terms of carbohydrates, it is well understood that nondigestible carbohydrates, which include dietary fibers, are referred to as “microbiome-accessible carbohydrates” and have a significant impact on the gut microbiome [65]. On the other hand, simple and digestible carbohydrates (e.g., sucrose, fructose, and glucose) are also indicated to modulate microbial diversity. Furthermore, they would impair gut permeability [68,69]. Fajstova et al. showed that a diet rich in simple sugars (high-sugar diet or HSD) induced an alteration in gut microbial composition and inflammatory immune responses in mice and that the HDS-fed mice showed increased severity of experimental colitis [68]. In the field of autoimmune diseases, it has been reported that dietary free sugar intake was associated with the disease activity of systemic lupus erythematosus (SLE) [69], which indicates a possible role for sugar intake in SLE and other autoimmune rheumatic diseases. Thus, as described by Anhê et al. [70], the relationship between blood glucose and gut microbes is bidirectional. In addition, the long-term effect of consuming a certain level of glucose would be a key focus of future research.

Of note, glucose also modulates the microbiome in the oral cavity. Dietary intake of sugar or carbohydrate can determine oral health conditions. The high glucose level in the oral cavity is suggested to induce inflammatory responses and alter the gingival or salivary microbiome [71,72]. Patients with diabetes have increased periodontitis risk and/or severity with an altered subgingival and salivary microbiome [71,73].

In patients with RA, oral health is focused on as an essential contributor to the disease. Specifically, gingivitis is shown as a risk factor to evoke the production of ACPA and further the RA occurrence [74]. In this regard, a potential role of the pathogenic bacteria, *Porphyromonas gingivalis*, in RA has been proposed since the bacteria can activate the peptidyl arginine deiminase (PADI) that can citrullinate proteins leading to an ACPA production [74,75]. On the other hand, therapy for periodontitis ameliorated the RA severity [76]. The patients with RA are therefore recommended to keep their oral condition, more specifically balanced oral microbiome, by controlling the glucose level both in serum and salivary appropriate to avoid excessive inflammatory responses.

## 6. Conclusions

Glucose is the essential and most crucial source of energy and homeostasis. It could also work as a key player in RA (Figure 1). Furthermore, it is also in a broad range of the pathophysiology of autoimmune disease through modulating immune responses and autoantigen expressions.

There is generally no need to restrict the sugar intake for patients with RA severely. However, the patients are advised to control their metabolic balance, including glucose levels, appropriately, to avoid the deteriorating excess glucose effect in chronic inflammation, which would create a vicious cycle to worsen the metabolic process and vice versa. Patients taking glucocorticoids or tacrolimus should be aware of their glucose level-elevating effect. Metabolic stability and a nonobese condition with maintained health in the oral cavity would benefit all patients with RA.

The investigations into glucose and its metabolism may pave the way for a better understanding of how carbohydrates contribute to RA occurrence and duration and other autoimmune diseases.

## Figures and Tables

**Figure 1 nutrients-14-02349-f001:**
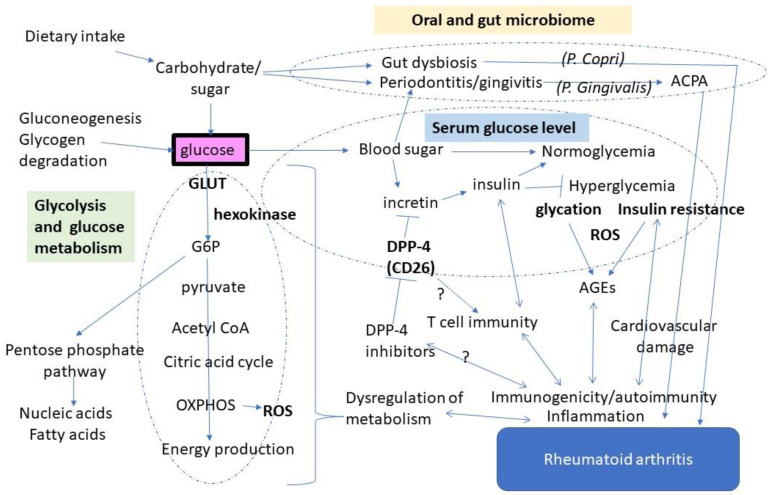
The schematic glucose demonstration as a center of the pathophysiology of rheumatoid arthritis. Glucose may play a role in the pathophysiology of autoimmunity and rheumatoid arthritis via its metabolic altera-tion, dysregulation of glucose levels, and influence in the inflammation in the oral cavity. G6P: Glycose-6-phosphate, OXPHOS: oxidative phosphorylation, P. copri: Prevotella copri, P. gingivalis: Porphyromonas gingivalis, DPP-4: dipeptidyl peptidase-4, AGEs: advanced glycation end-products, ACPA: anticitrullinated protein antibodies.

## Data Availability

Not applicable.

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
