# Peer review of "Glucose as a Potential Key to Fuel Inflammation in Rheumatoid Arthritis"

_nutrients, 2022, doi:10.3390/nu14112349_

Round 1

Reviewer 1 Report

The manuscript titled “Glucose as a potential key to fuel inflammation in rheumatoid arthritis” by Kayo Masuko discusses the current understanding on relationship between glucose, glucose metabolism and inflammation, and its role in pathophysiology of rheumatoid arthritis. A few suggestions to the authors:

1. The direct evidence linking glucose to rheumatoid arthritis is missing. The manuscript majorly discusses altered cellular metabolism in rheumatoid arthritis.

2. The relationship between glucose and inflammation is unclear. Whether elevated glucose levels drive inflammation or inflammation and hypoxic conditions in rheumatoid arthritis drive increased uptake of glucose and altered glucose metabolism is not entirely clear in the manuscript. Lines 56-60 present contradictory findings.

3. Authors should briefly discuss how glucose/glucose metabolism contributes to post translational modification of biomolecules resulting in autoimmunity.

4. Authors discuss DPP-4 as the link between diabetes and autoimmunity which may be working through insulin signaling and inflammation (unclear), where does glucose fit into the picture?

Author Response

Thank you for your detailed assessment and valuable comments on my manuscript.

  1. I agree with the reviewer that there is no direct evidence to link glucose to rheumatoid arthritis as the pathogenic agent. As the reviewer pointed out, current research interests have been accumulating in the field of altered cellular metabolism, in particular, of glucose; therefore, the issue has become the main focus of the manuscript.
  2. Thank you for the instructive comment. To support the suggested relationship between hyperglycemia and inflammation, and address the issue in lines 56-60 on page 2 more clearly, I added a paragraph describing the reported implication of hyperglycemia in inflammation via reactive oxygen species on page 2 in the revised manuscript with three new references.
  3. As suggested by the reviewer, there is no direct evidence to show that glucose would be a triggering factor for autoimmunity. However, as advanced glycation end products, a product of abnormal glucose metabolism and hyperglycemia, are demonstrated to be immunogenic, I reconstructed paragraph #3 (lines 97-99) to describe the issue more clearly.
  4. Thank you for the instructive comment. To clearly show the standpoint of glucose in the link between diabetes and rheumatoid arthritis, I added a figure (figure 1).

Reviewer 2 Report

are clarifying that glucose metabolism might be altered in rheumatoid arthritis (RA), which would play a role in the inflammatory phenotype of rheumatoid synovial fibroblasts. It may also play a role in a variety of autoimmune disease pathophysiology by modulating immune responses and modifying autoantigen expressions. Research into glucose and its metabolism could lead to a better understanding of how carbohydrates contribute to the occurrence and duration of rheumatoid arthritis and other autoimmune diseases. In this manuscript, the author's goal is to introduce recent topics concerning the role of glucose in RA. Overall, this is an interesting work.

1.    Authors should carefully check the writing and format before submitting the paper, and many faults should be modified in reference.

2.    There are some grammatical and spelling errors in the manuscript. Authors should carefully proofread their manuscripts.

3.    The author should cite some paper about rheumatoid arthritis: Biomaterials, 239, 2020, 119851; Biomaterials, 277, 2021, 121088

Author Response

Thank you for your favourable comment.

  1. Authors should carefully check the writing and format before submitting the paper, and many faults should be modified in reference.
  2. There are some grammatical and spelling errors in the manuscript. Authors should carefully proofread their manuscripts.

Thank you for the instruction. The format of the manuscript has been now set to the MDPI format. Also, the manuscript was checked by a native English-speaking editor.

  1. The author should cite some paper about rheumatoid arthritis: Biomaterials, 239, 2020, 119851; Biomaterials, 277, 2021, 121088

Thank you for your instruction. I have read the two papers; “PRP-chitosan thermoresponsive hydrogel combined with black phosphorus nanosheets as injectable biomaterial for biotherapy and phototherapy treatment of rheumatoid arthritis “ by Pan et al., and “A multifunctional nano-therapeutic platform based on octahedral yolk-shell Au NR`CuS: Phototermal/photodynamic and targeted drug delivery tri-combined therapy for rheumatoid arthritis” by Huang et al. The papers were exciting. However, the context does not seem to directly link with the current “opinion” paper concerning glucose metabolism; therefore, I would cite the references in more suitable manuscripts in future submissions. Thank you again for the valuable information.

Reviewer 3 Report

It would be of great inteterest for the readers when the author adds  a paragraph to the possible role of gingival bacteria and paraodontitis,  inducing anti CCP antibodies, their being promoted by sugar and smoking and consequent increasing activity of RA.

Can the author philosophize about the consequences regarding sugar use of treatments in RA? Corticosteroids?  MTX? Jak Ab? TNF alfa inhibitors?

Also I would suggest a conclusion and advise to the doctors treating RA patients. Is it advisible to omit eating glucose ? what can a patient await.

I wonder whether the Diabetes an Insulin part can be shortened as it appears not to play such an important role in RA.

Author Response

I appreciate the reviewer for the key instruction. I have added a paragraph describing the potential relationship between the intake of sugar or carbohydrate, periodontitis, and RA in the revised manuscript (lines 193 to 206).

Can the author philosophize about the consequences regarding sugar use of treatments in RA? Corticosteroids?  MTX? Jak Ab? TNF alfa inhibitors?

Also I would suggest a conclusion and advise to the doctors treating RA patients. Is it advisible to omit eating glucose ? what can a patient await.

Thank you for the suggestion. I added a summarized comment in the conclusion (lines 213 to 218).

I wonder whether the Diabetes an Insulin part can be shortened as it appears not to play such an important role in RA.

Thank you for your advice. I shortened the sentences regarding insulin. (section 4)

Round 2

Reviewer 1 Report

The authors have answered the questions satisfactorily.